# The Protein Corona Paradox: Challenges in Achieving True Biomimetics in Nanomedicines

**DOI:** 10.3390/biomimetics10050276

**Published:** 2025-04-29

**Authors:** Nicole M. Mayordomo, Ane Zatarain-Beraza, Fabio Valerio, Victoria Álvarez-Méndez, Paula Turegano, Lucía Herranz-García, Amaia López de Aguileta, Nicolas Cattani, Ana Álvarez-Alonso, Mónica L. Fanarraga

**Affiliations:** 1Molecular Biology Department, Universidad de Cantabria, Avda. Herrera Oria s/n, 39011 Santander, Spain; 2Grupo de Nanomedicina, Instituto de Investigación Valdecilla-IDIVAL, Avda. Herrera Oria s/n, 39011 Santander, Spain

**Keywords:** biocorona, nanoparticles, protein corona, protein interaction, biodistribution, conformational changes, nanoparticle design

## Abstract

Nanoparticles introduced into biological environments rapidly acquire a coating of biomolecules, forming a biocorona that dictates their biological fate. Among these biomolecules, proteins play a key role, but their interaction with nanoparticles during the adsorption process often leads to unfolding and functional loss. Evidence suggests that protein denaturation within the biocorona alters cellular recognition, signaling pathways, and immune responses, with significant implications for nanomedicine and nanotoxicology. This review explores the dynamic nature of the protein corona, emphasizing the influence of the local biological milieu on its stability. We synthesize findings from studies examining the physicochemical properties of nanoparticles—such as surface charge, hydrophobicity, and curvature—that contribute to protein structural perturbations. Understanding the factors governing protein stability on nanoparticle surfaces is essential for designing nanomaterials with improved targeting, biocompatibility, and controlled biological interactions. This review underscores the importance of preserving protein conformational integrity in the development of nanoparticles for biomedical applications.

## 1. Introduction

Nanoparticles (NPs) are transformative tools in nanomedicine, driving advancements in diagnostics, targeted drug delivery, and hyperthermia therapies. Their unique physicochemical properties, particularly their high surface-area-to-volume ratio, grant them exceptional surface reactivity, enabling rapid and context-specific interactions with biomolecules. Upon entering biological environments such as blood plasma, interstitial fluid, or intracellular spaces, NPs are rapidly coated by a dynamic layer of proteins and other biomolecules, collectively known as the “biomolecular corona” [1]. The composition of this corona varies according to the physicochemical properties of the NPs and the specific biological milieu, resulting in distinct biological outcomes.

The formation of the protein corona fundamentally alters the identity and behavior of NPs in vivo. It influences critical factors such as biodistribution, cellular uptake, and therapeutic efficacy by acting as a functional “camouflage” [2,3,4,5,6,7]. This biomolecular coating enables NPs to mimic biological structures, enhancing their biocompatibility and extending their circulation time. However, it also significantly affects their interactions with cells and the immune system, thereby shaping their overall therapeutic performance [8,9,10,11,12,13]. Given its pivotal role, understanding and controlling NP-protein interactions has become a central focus in nanomedicine.

Advances in molecular biology and proteomics have enabled researchers to engineer tailored coronas, thus offering immense potential to optimize NP performance, paving the way for the precise design of nanomaterials with reliable therapeutic or diagnostic functions. These innovations promise not only safer and more effective nanomedicines but also valuable insights into NP behavior within biological systems. The capability to design biomolecular coronas “à la carte” represents a groundbreaking step toward adaptive, context-sensitive nanomedicines. Such developments hold the potential to address critical challenges in personalized medicine, immune modulation, and targeted therapies, shaping the future of nanomedicine.

This review focuses on the molecular biology of the proteinaceous biomolecular corona, exploring the mechanisms that drive its formation, the physicochemical and biological factors that shape its nature, its composition upon contact with the entry pathway and in various biological media, and its profound influence on NP behavior in living systems. By integrating recent advances in the field, this work aims to provide a comprehensive perspective on corona dynamics and how a deeper understanding of these processes can enable the rational design of tailored biomimetic NPs. Such insights promise to advance nanomedicine by opening up new possibilities for precise, context-sensitive therapeutic and diagnostic applications.

## 2. Challenges in Understanding and Investigating the NP Corona

### 2.1. Formation and Basic Structure of the Biocorna

The formation of the biocorona, first observed by Vroman in the 1960s [14], involves the rapid adsorption of biomolecules onto the surface of NPs upon contact with biological fluids. This biomolecular layer represents the initial interface between NPs and their biological environment. Although the composition of the biocorona is highly variable and dependent on multiple factors that will be discussed in later sections, general principles govern its structure, dynamics, and effects on NP behavior in biological systems.

Structurally, the biocorona is divided into two main layers: the hard corona and the soft corona, depending on the stability of the adsorbed biomolecules. The hard corona is composed of biomolecules strongly adsorbed to the NP surface through electrostatic and hydrophobic interactions [15]. This layer is stable and remains attached for extended periods, even in dynamic biological environments. Its robustness allows for characterization using techniques such as centrifugation and washing, which remove weakly associated biomolecules without disrupting the hard corona.

Surrounding the hard corona is a more dynamic layer known as the soft corona [16]. It consists of biomolecules weakly associated with the NP surface and with each other through reversible interactions. Due to its transient nature, the composition of the soft corona fluctuates rapidly in response to changes in the biological environment. The biomolecules comprising this layer often dissociate during centrifugation or washing processes, making their study challenging [15].

### 2.2. The Dynamic Nature of the NP-Bio Interface

The biocorona of NPs is highly dynamic, undergoing constant evolution and re-equilibration as NPs transition through different biological fluids. This process involves the continuous exchange of biomolecules on the NP surface, with new biomolecules replacing those initially adsorbed during synthesis (Figure 1). Such dynamic restructuring allows the NP to adapt its surface properties to the prevailing biological environment, thereby acquiring a continually evolving biological identity.

The dynamic nature of the protein corona is influenced by a complex interplay of factors, including NP physicochemical properties, the surrounding biological environment, and temporal changes. These aspects will be further explored in the following sections.

Silica NPs(SiO_2_) exhibit rapid biocorona stabilization, reaching equilibrium within one hour. This behavior is driven by the preferential adsorption of small, highly mobile proteins like albumin, which are quickly displaced by proteins with higher binding affinities [14]. In contrast, hydrophobic NPs with apolar surfaces preferentially interact with a different set of proteins; for example, lipid NPs predominantly interact with biomolecules such as ApoE [17] and show a slower protein corona evolution [18]. Interestingly, some studies indicate that hydrophobic NPs can adsorb twice as much protein and form a more stable and less dynamic protein corona [19].

In general, hydrophobic/apolar surfaces enable stronger and more stable interactions with adsorbed proteins [14]. This occurs because proteins unfold when exposed to hydrophobic NPs, exposing their hydrophobic cores undergoing denaturation. This denaturation results in more stable protein-NP interactions, delaying the dynamic exchange characteristic of the Vroman effect (Section 4.3) and leading to a more persistent and less dynamic biocorona [18].

Interestingly, the biocorona acts not just as a dynamic interface but also as a molecular ‘memory’ of the NP’s journey through the body. Proteins in the hard corona preserve traces of prior encounters with various biological environments. This offers crucial insights into the NP’s behavior and its interplay with different biofluids.

### 2.3. Challenges in the Study of the Biocorona

Accurately determining the qualitative and quantitative composition of the biocorona is essential for understanding NP interactions with biological systems and optimizing their design for therapeutic or diagnostic applications. However, studying this molecular coating presents significant challenges due to its highly dynamic nature, requiring robust and reproducible isolation techniques to capture a representative snapshot at specific time points and within defined biological media [20,21,22].

One of the major challenges in biocorona research lies in the impact of methodological variations. Differences in NP dispersion, isolation protocols, and analytical techniques can significantly influence the observed composition of the biocorona [23,24]. Current isolation methods—such as centrifugation, precipitation, and magnetic separation followed by washing steps—tend to disrupt the biocorona by removing loosely bound molecules, particularly those in the soft corona. This often results in an incomplete representation of the native structure and dynamics of the biocorona [13,21,25].

To address these limitations, alternative techniques have been developed to better capture the transient nature of the biocorona. Methods such as transmission electron microscopy (TEM), dynamic light scattering (DLS), isothermal titration calorimetry (ITC), and mass spectrometry (MS) have proved useful. In addition, advanced approaches such as cryogenic electron microscopy, X-ray absorption near-edge structure (XANES) and circular dichroism allow more detailed and refined characterization of the biocorona. [20,21,25,26,27].

Another challenge in the study of the biocorona is the dominance of highly abundant proteins, such as albumin and immunoglobulins in serum, which often mask the detection of less abundant but potentially more significant proteins at the biological level. This limitation underscores the need for advanced analytical methods capable of preserving and identifying low-abundance biomolecules during the analysis [24]. This leads to fragmented and often contradictory findings on the true composition of biocoronas in different biofluids, hindering our understanding of this important biomolecular layer.

### 2.4. (Bio)Chemical Components of the Biocorona in Nanomedicines

Although often referred to as the “protein corona”, this terminology oversimplifies the biocorona’s molecular complexity. The biocorona consists of a diverse range of biomolecules, including proteins, lipids, nucleic acids, metabolites, carbohydrates, and other small molecules. These components collectively define the biocorona’s biological identity, profoundly influencing NP behavior, biodistribution, stability, and interactions within the biological milieu.

Lipids comprise a diverse group of compounds, including fatty acids, phospholipids, sterols, terpenes, and others, that play a fundamental role in the biocorona. Through hydrophobic interactions, these lipids contribute to the biocorona’s structural stability and functional versatility, enhancing its biological mimicry [28,29,30,31]. Notably, phospholipids, cholesterol, and fatty acids are particularly integral to the biocorona in systems like iron oxide NPs [31]. Cholesterol and triglycerides also exhibit strong binding affinity to polystyrene NPs exposed to serum [20].

In liposomal systems, the adsorption of plasma lipoproteins onto their surfaces introduces a broad spectrum of lipids, including phospholipids, steroids, carnitines, fatty alcohols, diglycerides, and fatty acids, resulting in a lipid-rich corona [32]. Similarly, lipid NPs frequently form biocoronas enriched with high-density lipoprotein (HDL), which has been demonstrated to be a superior predictor of in vivo activity compared to other conventional corona biomarkers, such as apolipoprotein E [17,33].

Importantly, the lipid composition of the biocorona can exhibit significant variability under different physiological conditions. Studies on carbon nanomaterials have demonstrated that distinct serum lipid profiles exert a profound influence on the protein composition of the biocorona [34]. This may significantly impact NP behavior in vivo, particularly in obese individuals where lipid profiles are markedly altered [35].

Nucleic acids are polyanionic polymers that play a central role in life, carrying genetic information and guiding protein synthesis. These molecules, including DNA and RNA in their different forms, are negatively charged due to their phosphate backbone, which drives interactions with NPs. The adsorption of nucleic acids onto NP surfaces primarily occurs through hydrogen bonding and electrostatic interactions mediated by these negatively charged phosphate groups [36]. Furthermore, specific types of NPs, such as gold NPs and carbon-based nanomaterials, exhibit strong interactions with nucleic acids through π–π stacking interactions [37,38]. In serum, both DNA and RNA are naturally present, with concentrations ranging from 1 to 35 ng/mL for DNA and approximately 2 ng/mL for RNA in healthy individuals [39].

Carbohydrates, while exhibiting lower adsorption rates compared to proteins or lipids, play a crucial role in carbohydrate-rich environments [31]. These biomolecules are fundamental to various biological interactions, particularly in cellular recognition and signaling. They are abundantly present on cell surfaces as glycoproteins and glycolipids, where they participate in a wide range of phenomena, including adhesion to epithelial surfaces and interactions with immune cells. The presence of carbohydrates within the corona can facilitate specific interactions with cell surface receptors, enhancing targeted delivery and therapeutic efficacy [40].

Carbohydrates can act as recognition molecules in infection and immunity processes. Sialic acid, a critical monosaccharide involved in numerous physiological processes, is notably abundant in plasma biocoronas [31]. Glycans associated with proteins, such as sialic acid, are also frequently found on NP surfaces, influencing interactions with immune cells. Furthermore, protein-carbohydrate interactions are crucial in numerous biological processes and are critically involved in the development of diseases such as cancer.

In food-related contexts, such as when NPs are used as food additives, sugars readily adsorb onto NP surfaces. The extent of sugar adsorption is influenced by the initial sugar concentration in the environment. For example, lactose adsorption on ZnO NPs can range from 8% to 37%, depending on the surrounding medium [31,41].

Finally, proteins in the corona play a pivotal role in determining their biological interactions and stability. Proteins are nanometric, highly complex molecules that are fundamental to life, performing an extraordinary diversity of functions that depend on their unique structures. Unlike other biomolecules, proteins possess a unique capability to interact specifically and selectively with biological targets. These interactions can significantly influence the biodistribution, cellular uptake, and immune response to nanomaterials. Furthermore, the protein corona can enhance the biocompatibility of nanomaterials, providing a ‘biological identity’ that aids in avoiding rapid clearance by the immune system. Conversely, there are certain proteins, such as apolipoprotein E, that facilitate NP targeting. For instance, apolipoprotein E assists in NP transcytosis across the blood–brain barrier by interacting with lipoprotein receptors on endothelial cells [42]. Consequently, the presence of proteins in the corona is crucial for the functionalization and effective application of nanomaterials in biomedical fields, surpassing the contributions of other biomolecules.

In addition to proteins, lipids, nucleic acids and carbohydrates, the biocorona of NPs can also include other biomolecules such as metabolites, small organic molecules or peptides [15]. Due to their importance, this review will concentrate on the protein composition of the biocorona, integrating findings from various studies to illuminate the general nature of the biocorona, how it continuously transforms, and its implication in biodistribution for the safety and efficacy of nanomedicines and nanodiagnostic agents.

## 3. The Influence of Nanomaterial Properties on the Protein Corona

Nanomaterial properties such as size, shape, surface charge, hydrophobicity, and surface roughness significantly influence the formation and dynamics of the biocorona [43,44]. These properties not only dictate the composition of adsorbed proteins but also impact NP biodistribution, cellular uptake, and biological interactions. This review will examine some of these properties and their critical importance.

### 3.1. The Shape

The shape of NPs directly influences protein adsorption and the subsequent biological responses (Table 1). Spherical NPs, due to their symmetrical geometry, form a uniform and stable protein corona. This even distribution of adsorbed proteins typically results in lower cellular uptake and reduced immune recognition, making spherical NPs suitable for long-circulating drug delivery systems [45]. Rod-shaped NPs, with their elongated structures, provide a larger surface area-to-volume ratio compared to spherical NPs. This increased surface area promotes greater protein adsorption, leading to a denser and more complex protein corona. Such properties enhance cellular uptake and targeting efficiency, especially in tumor-targeting applications, as rod-shaped NPs tend to align with cellular membranes during uptake [46]. Cubical NPs exhibit sharp edges and flat surfaces, leading to heterogeneous protein adsorption. The resultant irregular protein corona affects NP stability and cellular interactions, sometimes enhancing receptor-specific binding but reducing overall reproducibility [15,45]. More complex geometries, such as star-shaped or flower-like NPs, form highly irregular protein coronas. These structures can enhance specific interactions with cellular receptors, improving targeting capabilities. However, the irregular corona introduces variability in biological responses, complicating their application in reproducible therapeutic designs [15].

Specifically for gold NPs, García-Álvarez et al. [47] observed substantial variations in the protein composition of the biocorona surrounding gold nanorods compared to gold nanospheres. These differences were particularly pronounced for proteins involved in blood coagulation and immune responses.

### 3.2. The Size

NP size exerts a significant influence on biocorona properties, impacting protein adsorption, stability, and subsequent biological interactions (Figure 2). This phenomenon has been observed across a wide range of NP types, including latex, polymeric, ZnO, gold, TiO2, mesoporous silica, and solid lipid NPs [47,48,49,50].

Several published studies conclude that small NPs (1–10 nm) create dense, tightly bound protein coronas due to their high surface area-to-volume ratio. This results in high cellular uptake and the ability to penetrate deep into tissues, making small NPs ideal for intracellular delivery and tissue penetration [48]. Studies on NPs smaller than 10 nm reveal that the adsorbed proteins often dominate the size and shape of the NP-protein complex, shifting the biological interaction paradigm [51,52].

Medium-sized NPs (10–100 nm) form a more dynamic and less dense protein corona. The loosely bound proteins allow reversible interactions in biological environments, balancing stability and cellular uptake. This size range is optimal for drug delivery systems due to efficient biodistribution [46]. Larger NPs (100–300 nm) develop a heterogeneous and less dense protein corona. They tend to interact with a broader variety of proteins, leading to more complex biological responses. For NPs larger than 200 nm, the protein corona becomes sparse and highly heterogeneous, with weakly bound proteins (Figure 2). These particles are often used in applications requiring prolonged circulation times, such as diagnostic imaging or controlled drug delivery [16,48].

From a molecular biology perspective, when a protein interacts with an NP at a similar size scale, subsequent protein adsorption from the surrounding medium can lead to partial unfolding of the polypeptide. Studies have demonstrated this phenomenon, with gold nanorods serving as a notable example, inducing unfolding of bovine serum albumin (BSA) [53,54,55,56,57]. These protein conformational changes can significantly impact the properties and function of the adsorbed proteins, potentially exposing normally hidden residues. Consequently, this can significantly alter the overall behavior of the NP-protein complex within the biological environment. Later sections of this review will delve deeper into these aspects.

### 3.3. The Hydrophobicity

Hydrophobicity strongly influences protein–NP interactions (Table 1) [19,58]. Hydrophobic NPs tend to adsorb proteins via van der Waals forces or π–π interactions, often causing protein denaturation by exposing hydrophobic domains. Excessive hydrophobicity can lead to particle aggregation, reducing NP bioavailability [59,60,61,62]. Hydrophobic NPs also form a more stable and less dynamic protein corona [19]. This stability can influence NP biodistribution and reduce protein exchange rates, which might increase biocompatibility [63]. Conversely, hydrophilic NPs exhibit weaker protein adsorption and higher exchange rates, allowing for extended circulation times and lower immune recognition [19].

### 3.4. The Charge

It is assumed that the surface charge of NPs significantly influences the composition and properties of the protein corona (Table 1). Studies have shown that positively charged NPs tend to attract negatively charged plasma proteins, such as serum albumin and fibrinogen [64]. This suggests that the surface charge of NPs plays a crucial role in determining the type and binding affinity of adsorbed proteins, ultimately impacting the biological outcomes associated with the corona. Other studies have reported that protein decoration of negatively charged particles did not consistently correlate with protein size or charge, suggesting that electrostatic interactions alone may not be the sole driving force governingNP-protein interaction [48].

From a molecular biology perspective, protein charge, in addition to NP charge, significantly influences protein–NP interactions. Proteins can be positively or negatively charged depending on the pH of the environment and their isoelectric point (pI). Below their pI, proteins carry a net positive charge, while above it, they are negatively charged. This charge-based behavior is critical for understanding protein-NP functionalization. Consequently, anionic NPs preferentially adsorb proteins with a pI greater than 5.5, such as IgG, whereas cationic NPs favor proteins like albumin, which have a pI below 5.5 [65].

### 3.5. The Surface Roughness

NP surface roughness is another key factor in biocorona formation (Table 1). Smooth surfaces tend to adsorb more protein, resulting in thicker protein coronae, thus enhancing biological interactions. In contrast, rough-surfaced NPs have been shown to have a “biorepellent” effect, reducing non-specific protein adsorption [66]. These corona differences significantly influence cellular uptake, with smooth NPs being internalized more efficiently by various cell types, including macrophages and cancer cells [66].

### 3.6. The Chirality

In chemistry, chirality refers to the property of a molecule or ion that prevents it from being superimposed on its mirror image through any combination of rotations, translations, or conformational changes. This characteristic, fundamental across various scientific disciplines, has gained prominence due to discoveries such as intense polarization rotation in NPs (Table 1) [67].

Chirality has been shown to significantly influence the interactions and behaviors of NPs [59,60,61,62]. For instance, studies on serum albumin adsorption onto L- and D-chiral gold NPs demonstrated substantial differences in thermodynamics, adsorption orientation, and affinity [68]. Similarly, research on gold NPs functionalized with d-, l-, and racemic penicillamine found that the surface chirality of these NPs determines the orientation and conformation of transferrin, affecting its interaction with cellular receptors [69].

The impact of chirality extends to gold nanoclusters (NCs), where D-chiral NCs were shown to induce significant aggregation and activation of coagulation factor XII (FXII), whereas L-chiral NCs formed more stable bioconjugates with reduced autoactivation. These findings highlight the molecular mechanisms through which chirality modulates interactions with specific proteins [61].

In the case of carbon nanodots, chirality was observed to influence protein corona formation and cellular uptake, with differences exceeding 20%. Despite this, chirality did not affect key physicochemical properties such as fluorescence or colloidal stability, demonstrating the selective effects of chiral features on biological interactions [60]. Protein coronas on quantum dots (QDs) also showed strong surface chirality-dependent dynamics [62]. These findings collectively emphasize the profound role of chirality in NP-protein interactions.

**Table 1 biomimetics-10-00276-t001:** Properties of NPs and effects on the biocorona.

Features	Impact of the Protein Corona	References
Shape	Spherical NPs have the smallest surface. Rod/discoidal shapes have more surface area = stronger interactions.	[15,45]
Size	Affect the protein composition. Protein unfolding in small NPs.	[50,53,54,55,56,57]
Hydrophobicity	Stabilization of the protein corona but risk aggregation, reducing bioavailability. Causes protein denaturation.	[19,58,59,60,61,62,64,70]
Roughness	Surface roughness decreases protein absorption.	[64,66]
Charge	Positive charge enhances cellular uptake, and negative charge reduces uptake and prolongs circulation time.	[48]
Chirality	Influences protein adsorption, corona dynamics and biological outcomes.	[60,61,62,68,69]

## 4. Physicochemical Properties of the Local Milieu on Biocorona Formation

The physicochemical properties of the environment, such as ionic strength, pH, and temperature, significantly influence the formation and stability of the biocorona by modulating the adsorption of biomolecules onto the NP surface (Table 2). It is important to note that this section focuses on the impact of these intrinsic physicochemical properties and excludes the significant influence of variations in protein composition.

### 4.1. Ionic Strength

Ionic strength, defined as the concentration of ions in a solution, is a critical factor influencing the assembly, composition, and stability of the biocorona. It plays a crucial role in governing NP aggregation. In environments with high ionic strength, such as blood, plasma or phosphate-buffered saline (PBS), elevated ion concentrations (typically 150 mM ClNa) reduce electrostatic repulsion between similarly charged NPs and proteins, facilitating closer interactions between NPs and biomolecules, enabling robust protein adsorption onto the NP surface [15,71,72]. The increased ionic concentration enhances the retention of high-affinity proteins, leading to the formation of a denser and more stable protein corona [73]. This contributes to predictable and sustained biological responses, including immune evasion, prolonged circulation times, and improved therapeutic outcomes.

At high ionic strength, the reduced electrostatic repulsion between particles can lead to aggregation, diminishing the colloidal stability of NPs and potentially reducing their bioavailability and therapeutic efficacy. In biological media, ionic strength governs key intermolecular forces such as electrostatic interactions and van der Waals forces between NPs and biomolecules, thereby modulating the structural and functional properties of the biocorona [74,75].

Conversely, in low ionic strength environments, such as cerebrospinal fluid, weaker electrostatic screening results in higher protein exchange rates and a less stable biocorona. This increased dynamism leads to temporal changes in biocorona composition, which can significantly alter NP interactions with cells and tissues. The fluidity of the biocorona under these conditions may result in less predictable biological outcomes, posing challenges for applications requiring precise targeting or long-term circulation.

Magnetic iron oxide NPs provide a specific example of how ionic strength impacts protein corona stability. When varying the NaCl concentration in PBS, researchers observed that higher ionic strength reduced the stability of human serum albumin coatings on these NPs, negatively affecting their biocompatibility, circulation time, and overall performance in biomedical applications [74].

Interestingly, elevated ionic strength not only enhances protein adsorption but also increases particle-protein contact areas through mechanisms beyond simple electrostatic interactions. For instance, at high salt concentrations, the reduced repulsion between proteins and NPs increases the likelihood of hydrophobic interactions, further stabilizing the protein corona [70]. Studies from Cantarutti et al. [75] demonstrated that altering the ionic strength of the surrounding medium could transform the protein-NP interaction from a transient “soft corona”, characterized by rapidly exchanging proteins, to a more stable “hard corona”, where proteins exhibit longer residence times on the NP surface. Overall, the ionic strength of the surrounding medium is a powerful determinant of the biocorona’s properties, affecting NP behavior in diverse biological contexts.

### 4.2. Local pH

The pH of the surrounding environment significantly impacts the formation, composition, and stability of the protein corona on NP. Proteins are made up of amino acids, which have side chains that can gain or lose protons depending on the pH of their environment. This protonation/deprotonation leads to changes in the overall charge of the protein. As previously mentioned, a protein’s net charge is zero at its pI (Section 3.4). Below the pI, proteins carry a net positive charge, while above it, they are negatively charged. These charge variations significantly influence protein-NP interactions. Furthermore, pH-dependent changes in protein charge can induce conformational changes, potentially altering their structure and stability, which consequently affects their adsorption onto the NP surface and their subsequent interactions within the biological milieu.

Neutral pH environments, such as blood plasma (pH 7.4), promote the retention of proteins in their native conformation. This stability facilitates efficient binding to NPs and preserves natural protein–protein interactions, such as those involving cellular receptors [32,76]. Under these conditions, the protein corona is typically stable and predictable, enhancing the therapeutic potential of NPs by maintaining their biological compatibility.

In acidic environments, such as those found in tumor microenvironments or lysosomal compartments (pH 5–6), proteins undergo conformational changes that significantly affect their adsorption capacity and alter the biocorona composition. For instance, studies on iron oxide NPs demonstrate that maximum adsorption of proteins, fatty acids, and carbohydrates occurs at pH 4.0, where the acidic conditions facilitate strong interactions between biomolecules and the NP surface [31]. On the contrary, the stability of the adsorbed layer on iron oxide NPs diminishes as the pH increases from weakly acidic (pH 6.0–6.6) to slightly alkaline conditions (pH 7.5) [74].

Investigations into solid lipid NPs have demonstrated that the pH-dependent aggregation of protein-coated NPs directly affects their biological interactions. For instance, at pH 6.0, BSA corona formation causes solid lipid NPs to aggregate, which alters the protein’s secondary structure and reduces the cellular uptake of the NPs. In contrast, at neutral pH (7.4), solid lipid NPs remain dispersed, preserving their functionality and improving their stability for therapeutic applications [50].

### 4.3. Temperature

Temperature is a critical determinant of the kinetics and stability of protein adsorption onto NP surfaces. Physiological temperatures (35–41 °C) provide a balance between biocorona stability and dynamism, enabling controlled adsorption while preserving protein functionality essential for biomedical applications. At physiologically elevated temperatures (39–41 °C), such as those observed during fever, increased molecular kinetic energy accelerates protein adsorption and enhances biocorona dynamics with higher rates of protein exchange. Quantitative studies demonstrate that higher protein adsorption occurs on iron oxide NPs at these temperatures [31]. However, at 43 °C, proteins undergo conformational changes leading to denaturation. This denaturation results in the unfolding and expansion of protein polypeptides on the NP surface, which can reduce corona thickness. As will be reviewed in later sections, these changes significantly affect biological interactions [77].

Lower temperatures reduce molecular motion, stabilizing the protein corona by limiting protein exchange rates. At 13–23 °C, proteins such as albumin and apo-transferrin preferentially form monolayers on NP surfaces, stabilizing the corona structure [77]. Studies on carbon nanomaterials have revealed significant qualitative differences in biocorona composition between 4 °C and physiological temperatures [34]. These findings emphatically demonstrate the critical role of temperature in shaping the biocorona. Therefore, it is crucial to consider physiological conditions when optimizing biocorona characteristics for specific applications.

### 4.4. Time in Protein Biocorona Formation: The Vroman Effect

The duration of NP exposure to biological fluids is a critical determinant of the composition and stability of the protein biocorona. This phenomenon, termed the “Vroman effect” after its description by Leo Vroman, refers to the competitive adsorption of proteins from biofluids onto surfaces, including NPs. It is characterized by a sequential adsorption process where proteins with higher mobility and lower molecular weight initially dominate the surface, only to be displaced by proteins with stronger surface affinities and higher molecular weights. This dynamic pattern is also observed in larger biomaterial systems [14,78].

In biological fluids such as blood plasma, the high protein concentration intensifies competition for adsorption sites on the NP surface. Initially, smaller and more abundant proteins, like albumin, dominate the biocorona. Over time, as protein concentration increases, higher-affinity proteins such as IgG exhibit distinct dynamic adsorption kinetics, forming transient peaks within 24 h. Eventually, proteins like fibrinogen, with even greater affinity, displace earlier adsorbed proteins, resulting in a thinner and more stable corona [19,58]. This fibrinogen adsorption can also induce NP agglomeration, significantly affecting their stability and interactions with cells [19].

The molecular mechanisms underlying the Vroman effect on NPs Involve dynamic adsorption–desorption cycles and conformational changes in the adsorbed proteins. This process is driven by both competitive and cooperative interactions. For instance, smaller, mobile proteins such as albumin may reorganize within the corona, facilitating the subsequent adsorption of larger, higher-affinity proteins. This dynamic nature reflects the highly adaptable and transient characteristics of the biocorona. The transient complex model further elucidates this process, describing a three-step mechanism: (1) new proteins integrate into the adsorbed layer, (2) destabilize the existing protein configuration, and (3) eventually replace the earlier adsorbed proteins [79,80].

### 4.5. Reducing Conditions

Under reducing conditions, disulfide bonds within proteins, which are essential for maintaining their tertiary and quaternary structures, are cleaved. This cleavage leads to structural alterations, including the disassembly of protein complexes and the unfolding of polypeptide chains. As a result, hydrophobic regions that are typically buried in the native conformation become exposed. These redox-induced changes significantly impact protein functionality by altering epitopes, active sites, and ligand-binding domains, all critical for immune interactions, molecular binding, and overall protein function [81]. As discussed in later sections, such structural alterations profoundly affect the formation and stability of the protein corona on nanomaterials. Specifically, they influence immune protein adsorption, ultimately modulating the corona’s composition and dynamics [82,83].

Reducing environments are found in various tissues such as the liver, bone marrow, pancreatic β-cells, and placenta. At the intracellular level, reducing conditions are prevalent in compartments like the cytoplasm, lysosomes, nucleus, and mitochondrial matrix. These environments play a critical role in regulating redox states and contribute to the structural and functional plasticity of proteins interacting with nanomaterials.

### 4.6. Fluidics, Shear Stress

Flow dynamics, such as those present in blood flow, significantly influence the formation and composition of the protein corona on NPs. Under dynamic flow conditions, the continuous movement and shear forces lead to a more heterogeneous and dynamic protein corona compared to static conditions, which favor binding affinity for the NP surface. The varying flow rates and turbulence in the bloodstream cause different proteins to adsorb and desorb from the NP surface more frequently, resulting in a diverse and constantly changing protein corona [84]. In contrast, static conditions tend to produce a more stable and uniform protein corona, as the lack of movement allows proteins to bind more firmly and remain on the NP surface for longer periods. Some studies have demonstrated that protein coronas formed under flow conditions more accurately reflect in vivo environments, where NPs encounter a complex and dynamic biological milieu [85,86].

**Table 2 biomimetics-10-00276-t002:** Influence of the local milieu on NP biocorona.

Features	Impact of the Protein Corona	References
Ionic strength	Influences electrostatic interactions and van der Waals forces between NPs and biomolecules	[74,75]
pH	Affect NP surface charge and ionize functional groups on proteins, modifying adsorption	[31,50,74]
Temperature	Influence protein folding and corona dynamics. Lower temperatures reduce molecular motion, stabilizing the protein corona	[31,34,77]
Time	Changes in protein composition due to the Vroman effect	[78,79]
Redox state	Structural and stability changes in the protein corona	[82,83]
Fluidics	Flow dynamics (e.g., blood flow) create more heterogeneous PC compared to static conditions	[85,86]

## 5. The Molecular Changes Occurring in Proteins of the Corona

### 5.1. Protein Structural Changes Induced by Nanoparticles

Proteins are polymers composed of 20 distinct amino acids linked by peptide bonds, forming polypeptide chains. Their unique amino acid sequence determines both their three-dimensional structure and biological function. When folded into their functional “native” state, polypeptides adopt specific secondary and tertiary structures, which are critical for their activity.

Since protein structure dictates function, even minor disruptions in sequence or folding can profoundly alter the protein function. This principle is especially relevant in nanomaterial interactions, where adsorption onto NPs can induce structural changes—such as denaturation or unfolding—that compromise a protein’s ability to bind other molecules or perform its biological roles. Understanding these conformational shifts is thus essential for advancing nanomaterial applications in biological systems (Figure 3). Such changes depend on the protein’s structural flexibility and the physicochemical properties of the NP, including size, shape, surface charge, and coating [87,88,89,90].

Proteins in the “soft corona” typically maintain their native structure, whereas those in the “hard corona” often experience conformational changes (Table 3). Denaturation of the protein coating on NPs can have both beneficial and detrimental effects. While it may stabilize colloidal dispersions by preventing aggregation (see below), it can also compromise the intended biological targeting of NPs and trigger premature clearance by the immune system.

### 5.2. Factors That Trigger Protein Conformational Changes

Protein conformational changes induced by adsorption onto NP surfaces are influenced by a combination of factors. As previously discussed, these include both the properties of the surrounding medium (e.g., pH, temperature, reducing conditions) and the intrinsic physicochemical characteristics of the NP itself (e.g., size, morphology, surface charge, hydrophobicity). This interplay disrupts inter- and intramolecular interactions within the protein, leading to alterations in its secondary and tertiary structures.

The surface charge of NPs plays a critical role in protein conformational changes. For instance, oppositely charged NP surfaces and proteins often reduce alpha-helical content while increasing beta-sheet structures. This effect is particularly pronounced with metallic NPs due to their high surface charge densities. These interactions may exhibit cooperative or competitive effects, further modulating protein adsorption dynamics [46,89,90,91,92,93,94].

The size of NPs significantly influences the extent of structural alterations in adsorbed proteins. Smaller NPs, due to their higher curvature, induce more pronounced conformational changes compared to larger NPs, which display saturation behavior with reduced structural disruptions. For example, beta-lactoglobulin exhibits size-dependent conformational changes when adsorbed onto silica NPs of varying sizes (4 nm, 20 nm, and 100 nm). Larger particles, particularly under acidic conditions, lead to significant reductions in alpha-helical content [93]. Lysozyme also demonstrates size-dependent behavior when adsorbed onto silica NPs. Smaller NPs fail to reach adsorption saturation, whereas 20 nm NPs form monolayer biocoronas, and 100 nm NPs exhibit multilayer adsorption. These changes further diminish the alpha-helical content of lysozyme, significantly altering its structural integrity and functional properties [93,94,95,96].

The morphology of NPs also affects protein interactions. Spherical NPs, such as those made of gold and silver, show stronger interactions with proteins compared to nanoplates or nanorods. This difference is attributed to the higher curvature of spherical NPs, which enhances the density and intensity of interactions with adsorbed proteins [97]. For NPs smaller than 10 nm, adsorbed proteins dominate the size and shape of the resulting NP-protein complex, dictating its biological interactions [47,51,52].

**Table 3 biomimetics-10-00276-t003:** Examples of protein denaturation on NP surfaces.

Protein Type	Denaturation Evidence	References
Serum Albumin	Rapid conformational changes at both secondary and tertiary levels. Unfolding reported for SiO_2_, TiO_2_ and AuNRs	[53,54,55,57]
IgG	PS NP-induced structural and functional changes leading to aggregation. Activation of macrophage responses	[95,98,99,100,101]
Fibrinogen	Unfolding induced by negatively charged poly(acrylic acid)-conjugated Au NPs leading to aggregation	[99,102,103,104]
Transferrin	NPs surface chirality determines orientation and conformation	[69]
Lysozyme	SiO_2_ NPs trigger secondary structure alterations and reduced activity	[96,98,102,103,104]
Trypsin	Silica grades and temperatures impact its structure and function.	[105]
Beta-lactoglobulin	SiO_2_ NPs size and pH-dependent structural changes. Decreased alpha-helical content	[106]

### 5.3. Consequences of Conformational Changes After Protein Denaturation

The interaction of proteins with NPs can induce significant structural changes, often compromising their biological function [101]. This phenomenon can have detrimental consequences for the therapeutic efficacy of nanomedicines. NPs may fail to reach their intended target site due to protein adsorption, or they may elicit unintended immune responses, leading to their elimination before they can exert their therapeutic effect. In addition, the colloidal stability of NP depends on the biocorona, leading to the formation of insoluble fibrous aggregates called amyloids, which are associated with various pathologies [107].

Additionally, NP–protein complexes formed by small NPs (<10 nm) often assume protein-dominated structures, which further modulate biological interactions and immune responses [102]. For example, 4 nm gold NPs incubated with human serum formed protein-dominated complexes, where the proteins dictated the overall structure and biological interactions of the complex [52].

Protein adsorption onto NPs frequently induces conformational changes that modify their interactions with immune cells. These structural alterations can disrupt native protein function and expose cryptic epitopes, polypeptide sequences normally buried within the protein core [83,108,109,110]. The newly accessible epitopes may trigger aberrant immune responses and dysregulate cellular signaling pathways. Furthermore, such structural modifications can reveal previously hidden binding sites that activate immune recognition mechanisms, ultimately influencing the biodistribution and clearance profiles of the NPs within biological systems. This cascade of events underscores the complex interplay between the NP surface chemistry and protein corona formation in determining biological outcomes.

## 6. The Everchanging Nature of Protein Coronas: A Barrier to Effective Nanomedicine

### 6.1. Beyond the Blood: Diverse Coronas in the Human Body

The formation and composition of the protein corona on NPs vary significantly depending on the biological environment they encounter and the routes of entry into the body. NPs are exposed to diverse biological fluids during their journey through the body, each imprinting a unique biomolecular signature on the NP surface. For example, NPs inhaled into the respiratory tract interact with airway lining fluids, forming coronas dominated by surfactant proteins, albumin, and immunoglobulins, which influence pulmonary clearance and immune recognition [111,112]. Conversely, NPs ingested via food or water encounter gastrointestinal fluids rich in digestive enzymes like pepsin and pancreatin [113,114,115,116]. In the bloodstream, intravenously administered NPs are rapidly coated with serum proteins such as albumin, fibrinogen, and apolipoproteins, whereas in cerebrospinal fluid (CSF), low-abundance proteins like apolipoproteins and complement proteins dominate the corona composition [117,118,119,120]. NPs that enter via skin acquire a protein corona influenced by interactions with keratinocytes, sebum, immune proteins, and proteolytic enzymes present in the epidermis and dermis [121]. In summary, each biofluid imparts a unique protein fingerprint on the corona, with distinct functional consequences.

As NPs move between different biological environments in the body, the initial protein coronas continuously change during the transition between biofluids. However, proteins from previous contexts may persist, creating a “memory” effect. This continuous remodeling of the protein corona complicates the prediction of NP behavior and highlights the importance of entry routes. The variability underscores how the route of administration and local biofluid composition dictate NP interactions with biological systems.

### 6.2. The Impact of Individual Variability on Nanoparticle Protein Corona

Human physiological factors, including age, sex, diet, and health status, significantly influence protein corona formation. The unpredictability of the protein corona is emphasized by patient-specific variables such as biological sex, genetic ancestry, disease state, environment, and age, which impact the biodistribution, pharmacokinetics, cytotoxicity, and organ targeting of NPs [122]. This phenomenon has been extensively studied in carbon nanomaterials, where different normal human serum samples produced quantitatively and qualitatively distinct bicoronas [34]. These findings underscore the importance of considering individual variability in the development of personalized and effective nanomedicines.

### 6.3. The Need for Engineering Predictable Protein Coronas for Improved Nanomedicine Efficacy

Protein coronas function as biocamouflage, mediating biological interactions and determining the fate of NPs in vivo [123]. Importantly, the biological effects of NPs are dictated more by the composition of the protein corona than by the intrinsic properties of the NPs themselves. The spontaneous formation of protein coronas is inherently uncontrollable, driven by the unique molecular signature of each biological fluid NPs encounter. Without deliberate design of a tailored corona, this unpredictability leads to undesired interactions, inefficient targeting, and compromised biodistribution.

The variability becomes even more pronounced when NPs transition between tissues—such as from alveoli to blood or mucosa to circulation. These transitions remodel the corona as proteins from previous environments persist and combine with new ones, creating a “memory” effect. This dynamic nature renders predicting corona composition—and, by extension, NP behavior—virtually impossible.

Furthermore, individual physiological variability, including age, sex, diet, and health status, exacerbates this unpredictability, influencing corona formation, biodistribution, and NP functionality. Such variability highlights the necessity of designing biomimetic, durable, and controllable NP coatings. Engineered protein biocoatings can stabilize NPs, enhance targeting, and mitigate immune responses, addressing the challenges posed by spontaneous corona formation. 

## 7. Discussion and Conclusions

Throughout this review, we have focused our analysis on the changes occurring in the protein corona, although it is well known that other biomolecular components such as nucleic acids and lipids also play a role in the corona’s composition. Proteins, as active biomolecules, undergo functional modifications upon binding to NPs due to alterations in their three-dimensional structure. These structural changes are often influenced by the type, morphology, and size of NPs, as extensively discussed in this review.

When a protein undergoes conformational changes, it may lose its function. If the protein unfolds, it can expose novel epitopes that may be immunologically compromising. This structural alteration can render the entire NP susceptible to immune recognition. Once these exposed sequences are identified as “foreign”, the immune system initiates a cascade of responses, leading to the NP’s rapid clearance by macrophages. Consequently, this immunological reaction prevents NPs from fulfilling their intended role as targeted medical agents, significantly reducing their efficacy.

This review serves as a comprehensive overview of the dynamic interactions between proteins and NPs and highlights the need for strategic control over these interactions. A key challenge in nanomedicine is to prevent unwanted alterations in protein structure and immune system activation while maintaining the functional integrity of NPs.

To address this, an innovative approach is the rational design of artificial protein coronas tailored for specific biomedical applications. By engineering surface coatings composed of carefully selected or synthetically modified proteins, it is possible to minimize undesired conformational changes and immune responses. This biomimetic strategy aims to enable NPs to behave similarly to viruses, which have evolved highly efficient mechanisms for targeting and evading immune detection.

By leveraging bioengineering techniques, researchers can design NPs with precisely controlled protein coronas that facilitate selective targeting while avoiding rapid clearance by immune cells. This approach represents a promising frontier in nanomedicine, opening new possibilities for the development of highly efficient and specific drug delivery systems. Ultimately, mastering the control over protein corona formation will be a decisive step toward harnessing the full potential of NPs for therapeutic and diagnostic applications.

## Figures and Tables

**Figure 1 biomimetics-10-00276-f001:**
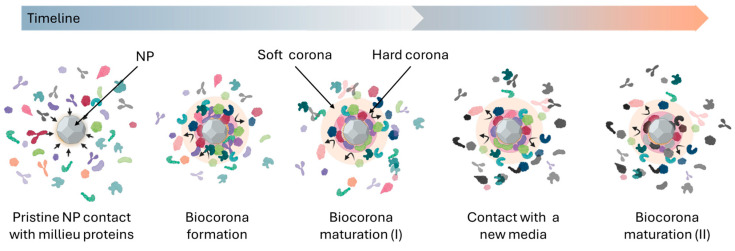
Dynamic process of biocorona formation and maturation on NPs. Upon introduction into a biological environment, pristine NPs rapidly adsorb a primary layer of biomolecules, forming the initial biocorona. This initial composition undergoes a dynamic maturation process over time, influenced by the NP’s encounter with different biological media containing varying protein compositions. This leads to the exchange of biomolecules and subsequent changes in the biocorona composition through successive maturation stages, ultimately resulting in a more stable and biologically relevant “mature” biocorona that dictates the NP’s interactions and fate within the biological system.

**Figure 2 biomimetics-10-00276-f002:**
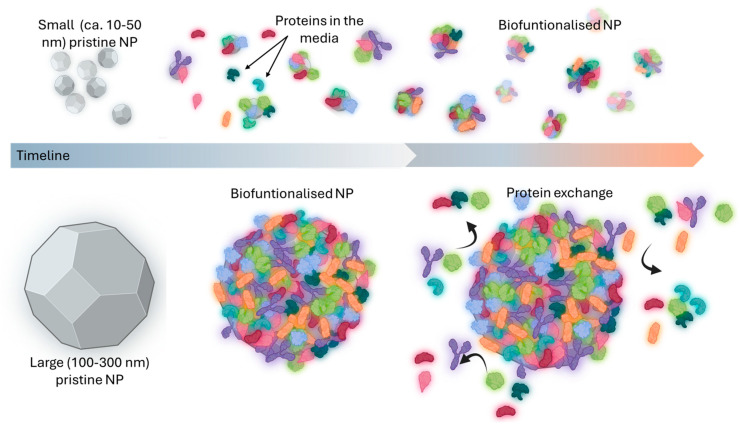
Size-dependent biocorona formation and maturation on NPs. This figure illustrates how the size of NPs influences both the initial formation and the subsequent maturation of the biocorona. It depicts potential differences in the relative amounts of biomolecules adsorbed, as well as the dynamic changes occurring over time (maturation stages) on NPs of varying dimensions.

**Figure 3 biomimetics-10-00276-f003:**
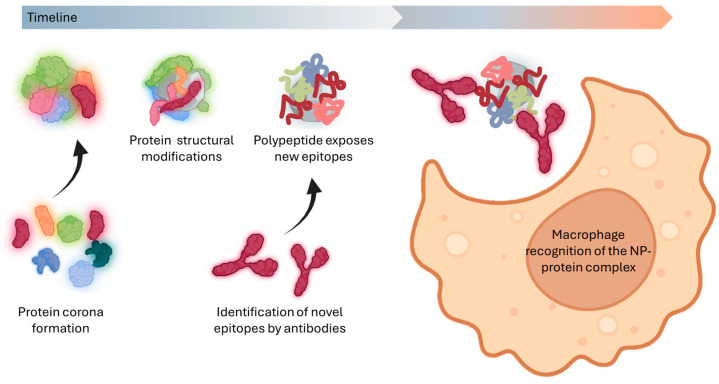
Biocorona protein denaturation and biological consequences. This illustration depicts the process of protein denaturation as polypeptides adsorb onto the surface of the NP. This structural alteration can lead to significant downstream effects, including the loss of specific targeting ligands and increased recognition by cells of the reticuloendothelial system (RES), ultimately affecting the NP’s biodistribution and efficacy.

## Data Availability

The original contributions presented in the study are included in the article, further inquiries can be directed to the corresponding authors.

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
