# Peer review of "The Protein Corona Paradox: Challenges in Achieving True Biomimetics in Nanomedicines"

_biomimetics, 2025, doi:10.3390/biomimetics10050276_

Round 1
Reviewer 1 Report
Comments and Suggestions for Authors
The authors correctly provided data about the nano-bio interactions in this work. I strongly suggest accepting this manuscript.
Author Response
REVIEWER #1:
Comment 1: The authors correctly provided data about the nano-bio interactions in this work. I strongly suggest accepting this manuscript.
Reply to reviewer. We are particularly grateful for Reviewer #1's strong endorsement and their specific acknowledgment of the data provided regarding nano-bio interactions.
Reviewer 2 Report
Comments and Suggestions for Authors
Comment 1: The abstract was well written, but the enhanced abstract should include more details about protein coronas and their biological importance to improve clarity and completeness.
Comment 2: The authors' introduction is well-structured. I suggest adding a discussion of how this study differs from previous research.
Comment 3: The figures should be enhanced with better clarity, labels, and captions for easier understanding.
Author Response
REVIEWER #2:
Comment 1: The abstract was well written, but the enhanced abstract should include more details about protein coronas and their biological importance to improve clarity and completeness.
Reply to reviewer. We have now written an enhanced abstract that reads this:
“Nanoparticles in biological environments rapidly acquire a protein-rich biocorona that dictates their fate, but this adsorption often denatures proteins, impacting cellular recognition, signaling, and immunity, crucial for nanomedicine and nanotoxicology. This review explores the dynamic protein corona, emphasizing the influence of the biological milieu and nanoparticle physicochemical properties (charge, hydrophobicity, curvature) on protein stability. Understanding these factors is essential for designing nanomaterials with improved targeting, biocompatibility, and controlled biological interactions, highlighting the need to preserve protein conformational integrity for safe and effective biomedical applications.”
Comment 2: The authors' introduction is well-structured. I suggest adding a discussion of how this study differs from previous research.
Reply to reviewer. This review distinguishes itself from previous works by placing a significant emphasis on the molecular biology of the protein corona. We delve into the reasons behind the structural changes that occur in adsorbed proteins and how these conformational alterations initiate a cascade of interactions with other proteins and cells, ultimately leading to the loss of targeting capabilities – a critical aspect often less detailed in prior reviews. While we appreciate the suggestion to further elaborate on this in the abstract, we believe the current wording, specifically the sentences "…Evidence suggests that protein denaturation within the biocorona alters cellular recognition, signaling pathways, and immune responses, with significant implications for nanomedicine and nanotoxicology. This review explores the dynamic nature of the protein corona, emphasizing the influence of the local biological milieu on its stability.”", already implicitly addresses this distinction by outlining the review's focus on the mechanisms and consequences of protein denaturation within the biocorona. For conciseness and to avoid redundancy within the limited space of the abstract, we feel this level of detail is appropriate. However, we are certainly open to revisiting the abstract in a subsequent revision round if the reviewer still deems further modification necessary. Thank you for your feedback.
Comment 3: The figures should be enhanced with better clarity, labels, and captions for easier understanding.
Reply to reviewer. We thank Reviewer #2 for the valuable suggestion regarding the clarity and labeling of the figures. In response, we have carefully revised both the figures (1&2) enhancing labels and visual clarity, and captions for figures 1,2&3 to improve overall understanding. Specifically, we have made the following modifications to the figure legends (all indicated in yellow in the final text of the manuscript):
Figure 1. Dynamic Process of Biocorona Formation and Maturation on Nanoparticles (NP). Upon introduction into a biological environment, pristine NPs rapidly adsorb a primary layer of biomolecules, forming the initial biocorona. This initial composition undergoes a dynamic maturation process over time, influenced by the nanoparticle's encounter with different biological media containing varying protein compositions. This leads to the exchange of biomolecules and subsequent changes in the biocorona composition through successive maturation stages, ultimately resulting in a more stable and biologically relevant "mature" biocorona that dictates the nanoparticle's interactions and fate within the biological system.
Figure 2. Size-Dependent Biocorona Formation and Maturation on NP. This figure illustrates how the size of nanoparticles influences both the initial formation and the subsequent maturation of the biocorona. It depicts potential differences in the relative amounts of biomolecules adsorbed, as well as the dynamic changes occurring over time (maturation stages) on NPs of varying dimensions.
Figure 3. Biocorona Protein Denaturation and Biological Consequences. This illustration depicts the process of protein denaturation as polypeptides adsorb onto the surface of the NP. This structural alteration can lead to significant downstream effects, including the loss of specific targeting ligands and increased recognition by cells of the reticuloendothelial system (RES), ultimately affecting the NP's biodistribution and efficacy.
We believe these enhancements to the figures and their legends will significantly improve the clarity and facilitate a better understanding of the concepts presented.
Reviewer 3 Report
Comments and Suggestions for Authors
Drug carriers in the form of nanoparticles adsorb biological molecules upon contact with physiological fluids to form a biocorona. The composition of the biocorona influences the properties of the nanoparticles, including their functionality, half-life, biodistribution, toxicity, and ability to avoid the immune system.
In my opinion, the topic of the paper is very interesting and well planned. The manuscript has been carefully prepared in terms of content and editing.
My comments:
1. In the literature section, an incomplete description of literature items numbered 16 and 131.
2. No citations for paragraphs:
- Line 361-366
- Line 470-474
3. In my opinion, the paragraph line131-137 on methods to enable more accurate characterisation of biocorona needs to be developed.
Author Response
REVIEWER #3:
Comment 1: In the literature section, an incomplete description of literature items numbered 16 and 131.
Reply to reviewer. We are deeply grateful to Reviewer #3 for his/her exceptionally thorough reading of our manuscript and for identifying these oversights, which were missed amidst the extensive references. Their keen attention to detail has been invaluable, and their contribution has significantly improved the overall quality of this work. References now read:
- Lundqvist, M.; Stigler, J.; Elia, G.; Lynch, I.; Cedervall, T.; Dawson, K.A. Nanoparticle Size and Surface Properties Determine the Protein Corona with Possible Implications for Biological Impacts; Proc Natl Acad Sci U S A 2008; 105:14265-70. doi: 10.1073/pnas.0805135105.
- Morshed, N.; Rennie, C.; Faria, M.; Collins-Praino, L.; Care, A. Protein Coronas Derived from Cerebrospinal Fluid Enhance the Interactions Between Nanoparticles and Brain Cells. bioRxiv, 2024, 2024.05. 31.596763, doi: 10.1101/2024.05.31.596763.
Comment 2: 2. No citations for paragraphs: Line 361-366. and- Line 470-474
Reply to reviewer. The revised texts now read:
- Line 361-366.
Ionic strength, defined as the concentration of ions in a solution, is a critical factor influencing the assembly, composition, and stability of the biocorona. It plays a crucial role in governing NP aggregation. At high ionic strength, the reduced electrostatic repulsion between particles can lead to aggregation, diminishing the colloidal stability of NPs and potentially reducing their bioavailability and therapeutic efficacy. In biological media, ionic strength governs key intermolecular forces such as electrostatic interactions and van der Waals forces between NPs and biomolecules, thereby modulating the structural and functional properties of the biocorona [78-79].
- Line 470-474 The paragraph has citations, and it reads:
The transient complex model further elucidates this process, describing a three-step mechanism: (1) new proteins integrate into the adsorbed layer, (2) destabilize the existing protein configuration, and (3) eventually replace the earlier adsorbed proteins [84,85].
Comment 3: In my opinion, the paragraph line131-137 on methods to enable more accurate characterisation of biocorona needs to be developed.
Reply to reviewer. Thank you for your comment. We acknowledge the importance of this aspect. The manuscript text already highlights, the methodological variations and inherent challenges in biocorona research, such as the disruptive nature of isolation protocols (centrifugation, precipitation, magnetic separation) leading to an incomplete representation of the native structure (as mentioned in references [13, 21, 25, 26]), and the dominance of highly abundant proteins masking less abundant ones (as noted in [24]), are significant considerations. However, the primary focus of this review is to explore the changes in the molecular biology of the biocorona throughout its maturation process and the subsequent biological implications, rather than providing an in-depth methodological study for biocorona characterization. While we have included a brief overview of techniques like TEM, DLS, ITC, MS, cryogenic electron microscopy, XANES, and circular dichroism (as detailed in references [20, 21, 26–28]), we believe the current extent of this section is sufficient within the scope of this review. Expanding it further to provide a detailed methodological analysis would shift the focus away from the core topic of the biocorona's molecular evolution and its biological consequences, such as loss of targeting. We trust that the current level of detail adequately supports the context of our discussion on the biological changes occurring during biocorona maturation.
Thank you again for your suggestion. We hope you understand our rationale for maintaining the current level of detail regarding the biocorona characterization methods. Given that the primary focus of this review is the molecular biology of the biocorona's maturation and its biological consequences, we believe the current scope of this section is appropriate and sufficient for the context of our discussion. We trust that this approach aligns with the overall aims of the review and hope this is acceptable.
Reviewer 4 Report
Comments and Suggestions for Authors
Dear Editor, in this review authors explore the dynamic nature of the protein corona, emphasizing the influence of the local biological milieu on its stability. Findings from studies examining the physicochemical properties of nanoparticles—such as surface charge, hydrophobicity, and curvature—that contribute to protein structural perturbations, have been presented. The review is well organized and contains some interesting results published in recent works. For this reason, I propose to accept it for publication.
Author Response
REVIEWER #4:
Comment 1: Dear Editor, in this review authors explore the dynamic nature of the protein corona, emphasizing the influence of the local biological milieu on its stability. Findings from studies examining the physicochemical properties of nanoparticles—such as surface charge, hydrophobicity, and curvature—that contribute to protein structural perturbations, have been presented. The review is well organized and contains some interesting results published in recent works. For this reason, I propose to accept it for publication.
Reply to reviewer. Dear Reviewer #4, we are deeply grateful for your positive assessment of our work and your recommendation for acceptance. Your comment, "The review is well organized and contains some interesting results published in recent works. For this reason, I propose to accept it for publication," is exceptionally encouraging and means a great deal to us, especially as a team with many early-career researchers. Thank you for recognizing the value and organization of our review.
Round 2
Reviewer 3 Report
Comments and Suggestions for Authors
Accept in present form